# Mild antecedent COVID-19 associated with symptom-specific post-acute sequelae

Tiffany A. Walker[1,2]*, Alex D. Truong[3], Aerica Summers[2], Adviteeya N. Dixit[3], Felicia C. Goldstein[4], Ihab Hajjar[1,4], Melvin R. Echols[5], Matthew C. Woodruff[6], Erica D. Lee[7], Seema Tekwani[3], Kelley Carroll[2], Ignacio Sanz[6], F. Eun-Hyung Lee[3], Jenny E. Han[2,3]

1 Department of Medicine, Division of General Internal Medicine, Emory University, Atlanta, GA, United States of America, 2 Grady Post-COVID Clinic, Grady Memorial Hospital, Atlanta, GA, United States of America, 3 Department of Medicine, Division of Pulmonary, Allergy, Critical Care and Sleep Medicine, Emory University, Atlanta, GA, United States of America, 4 Department of Neurology, Emory University School of Medicine, Atlanta, Georgia, United States of America, 5 Department of Cardiology, Morehouse School of Medicine, Atlanta, Georgia, United States of America, 6 Department of Medicine, Division of Rheumatology, Lowance Center for Human Immunology, Emory University, Atlanta, GA, United States of America, 7 Department of Psychiatry and Behavioral Sciences, Emory University School of Medicine, Atlanta, Georgia, United States of America

* Twalk25@emory.edu

**Data Availability Statement:** All relevant data are within the manuscript and its Supporting Information files.

## Abstract

### Background

The impact of COVID-19 severity on development of long-term sequelae remains unclear, and symptom courses are not well defined.

### Methods

This ambidirectional cohort study recruited adults with new or worsening symptoms lasting ≥3 weeks from confirmed SARS-CoV-2 infection between August 2020–December 2021. COVID-19 severity was defined as severe for those requiring hospitalization and mild for those not. Symptoms were collected using standardized questionnaires. Multivariable logistical regression estimated odds ratios (OR) and 95% confidence intervals (CI) for associations between clinical variables and symptoms.

### Results

Of 332 participants enrolled, median age was 52 years (IQR 42–62), 233 (70%) were female, and 172 (52%) were African American. Antecedent COVID-19 was mild in 171 (52%) and severe in 161 (48%). In adjusted models relative to severe cases, mild COVID-19 was associated with greater odds of fatigue (OR:1.83, CI:1.01–3.31), subjective cognitive impairment (OR:2.76, CI:1.53–5.00), headaches (OR:2.15, CI:1.05–4.44), and dizziness (OR:2.41, CI:1.18–4.92). Remdesivir treatment was associated with less fatigue (OR:0.47, CI:0.26–0.86) and fewer participants scoring >1.5 SD on PROMIS Cognitive scales (OR:0.43, CI:0.20–0.92). Fatigue and subjective cognitive impairment prevalence was higher 3–6 months after COVID-19 and persisted (fatigue OR:3.29, CI:2.08–5.20;

**Funding:** TAW and JEH were funded by the Woodruff Health Sciences Center COVID-19 CURE Award. There is no associated award number. http://whsc.emory.edu/research/covid-19-research/index.html The funders had no role in study design, data collection and analysis, decision to publish, or preparation of the manuscript.

**Competing interests:** I have read the journal's policy and the authors of this manuscript have the following competing interests: Dr. F. Eun-Hyung Lee reports grants from Genetech and the Gates Foundation. She has received royalties for BLI, Inc and consulting fees from Be Bio Pharma. She received honoraria from Gerontological Advanced Practice Nurses Association and has patents for plasma cell survival media and MENSA. She is the founder of the MicroB-plex, Inc. Dr. Ignacio Sanz reports grants from GlaxoSmithKline, Bristol Myers Squibb, Exagen, and consulting fees from Pfizer, Octagon, and Bristol Myers Squibb. He serves on the DSMB for and has stock options in Kyverna. This does not alter our adherence to PLOS ONE policies on sharing data and materials.

cognitive OR:2.62, CI:1.67–4.11). Headache was highest at 9–12 months (OR:5.80, CI:1.94–17.3).

## Conclusions

Mild antecedent COVID-19 was associated with highly prevalent symptoms, and those treated with remdesivir developed less fatigue and cognitive impairment. Sequelae had a delayed peak, ranging 3–12 months post infection, and many did not improve over time, underscoring the importance of targeted preventative measures.

## Background

The coronavirus disease 2019 (COVID-19) has resulted in over 685 million cases globally and resulted in greater than 6.8 million deaths [1]. Although there are numerous publications describing the wide range of acute phase manifestations [2], there are more limited data on post-acute sequelae of SARS-CoV-2 (PASC). Common symptoms include fatigue, dyspnea, cognitive impairment, and pain syndromes, but can span multiple organ systems [3–6]. While the incidence of PASC is unknown, symptoms lasting $\geq$ 3 months have been reported in 21–33% of COVID-19 survivors in prospective cohort studies [7–10]. PASC can significantly impair quality of life [3, 6, 11, 12], and in many, debilitating symptoms can persist for over a year with an unclear timeline for symptom resolution [3, 11, 13].

Identification of PASC risk factors and clinical course is essential to guiding clinical education and COVID-19 mitigation strategies. Although early studies showed higher rates of persistent symptoms in patients who experienced severe SARS-CoV-2 infections, the impact of antecedent COVID-19 severity on PASC development remains unclear as further studies have shown conflicting findings [3, 11, 14–17]. Delineation of PASC symptom trajectories may provide insight into potential pathophysiological processes. Given the heterogeneity of PASC symptoms and presentations, there are data to support distinct phenotypes that may result from different underlying mechanisms [6].

We identified risk factors associated with persistent symptoms in adults presenting to our PASC clinics, including association with acute COVID-19 severity and treatment. Additionally, although PASC has been defined by the Delphi consensus as symptoms usually lasting 3 months from COVID-19 onset [18], little is known about underlying mechanisms driving PASC phenotypes and the timeline in which these processes take course. Therefore, we characterized prevalent symptom profiles across 3-month intervals beginning 3 weeks from acute COVID-19 onset to inform symptom courses in our population.

## Methods

### Study population and design

This ambidirectional cohort study consecutively recruited adults aged $\geq$18 years with new or worsening symptoms lasting $\geq$3 weeks from confirmed SARS-CoV-2 test from two academic PASC clinics at Emory University and Grady Memorial Hospital in Atlanta, GA between August 2020–December 2021. We excluded participants who had 1) no persistent symptoms $\geq$3 weeks from disease onset, 2) no documentation of positive SARS-CoV-2 RNA or antigen from nasopharyngeal swab. Due to the high volume of patients referred to our PASC clinics during the study period, standardized follow-up could not be consistently accommodated. For

the purposes of this study, participants are only represented at one time point. Ethics approval was obtained from Emory University IRB prior to study initiation, and written consent was obtained as indicated.

## Data collection and case definitions

Sociodemographic (age, sex, self-reported race, and insurance status), comorbidities, and acute COVID-19 data (disease onset, severity, and treatments <3 weeks of disease onset) were collected by study physicians during the in-person visit and confirmed by electronic health record review. COVID-19 vaccination data were not collected systematically, so this data point was excluded from our analysis. Infections occurring from March 1, 2020-June 30, 2021 were defined as Alpha strain and those occurring from July 1-December 1, 2021 were defined as Delta variant. As outpatient oxygen saturation stratification was not consistently available for mild acute COVID-19 cases, we defined mild cases as acute COVID-19 not requiring in-patient hospitalization within 14 days of symptom onset and severe cases as requiring in-patient hospitalization or intensive care unit (ICU) admission for COVID-19 complications within 14 days of symptom onset. Standard of care treatment for severe acute COVID-19 with SpO2 ≤ 94% during the course of this study included remdesivir 200 mg IV loading dose on day 1 followed by 100 mg IV maintenance dose daily for 5 days and Dexamethasone 6 mg IV daily for 10 days or until hospital discharge. Remdesivir and corticosteroid regimens prescribed to mild acute COVID-19 participants did not follow a standardized protocol and occurred at the discretion of the clinical provider. In this case, remdesivir was typically administered as a 3-day infusion at local infusion centers. Mild acute COVID-19 patients with ≥ 1 risk factors were eligible for monoclonal antibody treatment beginning August 2021. PASC duration was quantified as months from incident COVID-19 onset and analyzed in 3-month intervals (0–3, 3–6, 6–9, 9–12, ≥12 months).

New symptoms or worsening symptoms since COVID-19 onset persisting ≥3 weeks were collected using a standardized review of systems with dichotomous scoring (presence/absence) and were confirmed by clinician interview (S1 Table). To estimate severity of the most prevalent PASC symptoms in our clinic, participants completed validated patient-reported outcome (PRO) tools including the PROMIS Cognitive Function 8A [19], PROMIS Dyspnea 10A [20], PROMIS Fatigue 7A [21], Patient Health Questionnaire (PHQ-9) [22], and Generalized Anxiety Disorder (GAD-7) scales [23]. Study data were collected and managed using REDCap electronic data capture tools hosted at Emory University [24].

## Statistical analysis

Sociodemographic and clinical characteristics are presented as absolute values and percentages for categorical values or medians and interquartile ranges (IQR) for continuous variables. Chi-square or Fisher's exact tests were calculated to assess differences in categorical variables between groups where appropriate. PROMIS Patient-reported outcome measures were analyzed by comparing the proportion of respondents scoring ≥1.5 standard deviations (SD) from the mean. PHQ-9 and GAD-7 were analyzed comparing the proportion of respondents scoring ≥10, indicating moderate-severe disease on population-based norms. A sample size calculation of 300 was determined to achieve >90% power to detect changes in sequelae trajectories in longitudinal analysis. Univariable logistic regression was conducted to estimate odds ratios (OR) and 95% confidence intervals (CI) for associations between PASC symptoms or PROs and PASC duration, COVID-19 variant, and COVID-19 severity. Multivariable logistic regression analysis was used to estimate OR and 95% CI for association between clinical variables (sociodemographic, comorbidity, acute COVID-19 severity and treatment, and PASC

duration) and PASC symptoms or PROs. Race and ethnicity were analyzed together as one variable in models. Variables that were significant on univariate analysis and those with clinical relevance were incorporated into the multivariable model. Akaike information criterion (AIC) was used to determine goodness-of-fit, and variables optimizing AIC were retained. Two-tailed p-values ≤0.05 were considered significant. We included participants for whom all variables of interest were available. No imputation was employed for missing data. Analysis was performed using SAS 9.4 [25].

## Results

### Study population

During the study period, 332 participants enrolled (Table 1). The median age was 52 years (IQR 42–62), 233 (70%) were female, and more than half were self-reported African American/Black (172, 52%), followed by Caucasian/White (103, 31%), Hispanic (15, 5%), Asian (13, 4%), American Indian/Alaskan (2, <1%), and (2, <1%) Pacific Islander/Native Hawaiian. Median PASC duration was 107 days (IQR 60–191), and 145 (44%) participants presented between 3 weeks-3 months from incident infection, 101 (30%) between 3–6 months, 42 (13%) between 6–9 months, 26 (8%) between 9–12 months, and 18 (5%) greater than 12 months. The median Charleston comorbidity index (CCI) was 2 (IQR 1–4). As a proxy for socioeconomic status, 110 (33%) were uninsured or Medicaid recipients. Antecedent COVID-19 infections were primarily Alpha strain (94%) and the remaining were Delta variant (S2 Table). COVID-19 severity was mild in 171 (52%) and severe in 161 (48%). Of the severe infections, median length of hospital stay was 8 days (IQR: 5–14), 40 (25%) were admitted to the intensive care unit (ICU), and mechanical ventilation was required in 18/40 (45%) of those admitted to the ICU.

### PASC risk factors

The most common symptoms were dyspnea (n = 236, 71%), fatigue (n = 199, 60%), subjective cognitive impairment (n = 156, 47%), cough (n = 98, 30%), dizziness (n = 73, 22%), and headache (n = 69, 21%) (Table 2). Severe fatigue (n = 76/263, 23%) and severe subjective cognitive impairment (n = 60/259, 18%) defined as scoring >1.5 SD on PROMIS scales were common, whereas severe dyspnea (n = 9/253, 3%) was infrequently reported. New or worsening moderate-severe depression and anxiety were seen in 26% and 16%, respectively. In adjusted models, mild acute COVID-19 severity was associated with higher rates of fatigue (OR: 1.83, 95% CI: 1.01–3.31), subjective cognitive impairment (OR: 2.76, 95% CI: 1.53–5.00), headaches (OR: 2.15, 95% CI: 1.05–4.44), and dizziness (OR: 2.41, 95% CI: 1.18–4.92) in our population (Fig 1). Additionally, they were less likely to demonstrate severe dyspnea, defined as scoring >1.5 SD on the PROMIS Dyspnea Scale (OR: 0.03, 95% CI: 0.002–0.47). No association was seen between COVID-19 severity and other reported symptoms. Similarly, PASC duration was positively correlated with fatigue (OR: 1.54, 95% CI: 1.22–1.94), subjective cognitive impairment (OR: 1.47, 95% CI: 1.19–1.81), headaches (OR: 1.50, 95% CI: 1.18–1.90), severe fatigue (OR: 1.73, 95% CI: 1.35–2.22), and severe subjective cognitive impairment (OR: 1.32, 95% CI: 1.01–1.72), indicating higher odds of these symptoms in patients with longer PASC durations.

Women reported more fatigue (OR: 1.83, 95% CI: 1.01–3.31), headaches (OR: 2.68, 95% CI: 1.26–5.71), and cognitive impairment (OR: 1.77, 95% CI: 1.04–3.00) compared with men, and the perceived cognitive impairment was five times more severe in women (OR: 5.65, 95% CI: 2.00–16.0). Advanced age (≥65 years) was modestly associated with fatigue (OR: 1.03, 95% CI: 1.00–1.06). Private insurance coverage (OR: 2.41, 95% CI: 1.94–3.00) and white race (OR: 1.61, 95% CI: 1.33–1.95) were associated with severe anxiety but had no effect on other symptoms.

**Table 1. Sociodemographic and clinical characteristics in participants with post-acute sequelae of SARS-CoV-2—August 2020–December 2021.**

| Characteristics | | Acute COVID-19 Severity | | Months after incident infection | | | | |
|---|---|---|---|---|---|---|---|---|
| | All participants | Mild | Severe | 0–3 | 3–6 | 6–9 | 9–12 | ≥12 |
| | N = 332 | N = 171 | N = 161 | N = 145 | N = 101 | N = 42 | N = 26 | N = 18 |
| | (column %) | (col %) | (col %) | (col %) | (col %) | (col %) | (col %) | (col %) |
| Age, median (IQR) | 52 (42–62) | 47 (39–56) | 59 (50–67) | 54 (41–64) | 53 (43–62) | 51 (47–59) | 51 (43–56) | 47 (39–64) |
| Female | 233 (70) | 132 (77) | 100 (63) | 100 (69) | 70 (69) | 33 (79) | 16 (62) | 14 (78) |
| Race | | | | | | | | |
| AI or AN | 2 (<1) | 1 (<1) | 1 (<1) | 2 (1) | 0 | 0 | 0 | 0 |
| Asian | 13 (4) | 7 (4) | 6 (4) | 5 (3) | 6 (6) | 2 (5) | 0 | 0 |
| Black or AA | 172 (52) | 68 (40) | 108 (65) | 86 (60) | 49 (49) | 21 (50) | 11 (42) | 5 (28) |
| Hispanic | 15 (5) | 6 (4) | 9 (6) | 5 (4) | 5 (5) | 1 (2) | 3 (12) | 1 (6) |
| NH or Other PI | 1 (<1) | 1 (<1) | 0 | 1 (<1) | 0 | 0 | 0 | 0 |
| White | 103 (31) | 66 (39) | 37 (23) | 38 (26) | 37 (36) | 13 (31) | 7 (27) | 8 (44) |
| Other/Unknown | 25 (8) | 22 (13) | 3 (2) | 7 (5) | 4 (4) | 5 (12) | 6 (19) | 4 (22) |
| Insurance | | | | | | | | |
| Uninsured | 74 (22) | 39 (23) | 35 (22) | 33 (45) | 20 (20) | 10 (24) | 8 (31) | 3 (17) |
| Medicaid | 38 (12) | 20 (12) | 18 (11) | 15 (10) | 11 (11) | 6 (15) | 3 (12) | 3 (17) |
| Medicare | 41 (12) | 8 (5) | 32 (20) | 17 (12) | 14 (14) | 5 (12) | 3 (12) | 2 (11) |
| Private | 172 (52) | 98 (58) | 74 (46) | 78 (54) | 52 (52) | 20 (49) | 12 (46) | 10 (56) |
| CCI, median (IQR) | 2 (1–4) | 2 (1–3) | 3 (2–5) | 3 (1–4) | 2 (1–4) | 3 (1–4) | 2 (1–4) | 2 (1–4) |
| Acute COVID-19 Severity | | | | | | | | |
| Asymptomatic | 1 (<1) | 1 (<1) | 0 | 0 | 1 (1) | 0 | 0 | 0 |
| Not hospitalized | 170 (51) | 170 (99) | 0 | 63 (44) | 55 (54) | 25 (60) | 17 (65) | 10 (56) |
| Hospitalized | 120 (36) | 0 | 120 (75) | 61 (42) | 33 (33) | 16 (38) | 5 (19) | 5 (28) |
| ICU | 40 (12) | 0 | 40 (25) | 20 (14) | 12 (12) | 1 (2) | 4 (15) | 3 (17) |
| Acute COVID-19 Treatment | | | | | | | | |
| Remdesivir | 114 (34) | 4 (2) | 110 (69) | 62 (43) | 28 (28) | 12 (29) | 8 (31) | 4 (22) |
| Corticosteroids | 184 (55) | 53 (31) | 131 (82) | 96 (66) | 51 (51) | 17 (41) | 11 (42) | 9 (50) |
| Anticoagulation | 128 (39) | 11 (6) | 117 (73) | 69 (48) | 34 (34) | 10 (24) | 9 (35) | 6 (33) |
| Monoclonal Antibodies | 18 (5) | 6 (4) | 12 (8) | 13 (9) | 3 (3) | 2 (5) | 0 | 0 |
| Plasma exchange | 7 (2) | 0 | 7 (4) | 6 (4) | 0 | 1(2) | 0 | 0 |
| PASC Duration, median days (IQR) | 107 (60–191) | 134 (65–209) | 90 (59–146) | 56 (45–71) | 121 (107–153) | 210 (193–230) | 310 (250–337) | 416 (385–463) |

IQR = Intraquartile range

AI = American Indian

AN = Alaska Native

AA = African American

NH = Native Hawaiian

PI = Pacific Islander

CCI = Charleston Comorbidity Index

COVID-19 = Coronavirus disease 2019

ICU = Intensive Care Unit

PASC = Post-acute sequelae of SARS-CoV-2

Those reporting cough were more likely to have received corticosteroids (OR: 2.15, 95% CI: 1.20–3.87) during acute infection. In subgroup analyses, the Alpha strain was associated with less anxiety (S2 Table) and critical COVID-19 was associated with development of more dyspnea and less subjective cognitive impairment, joint pain, anosmia, and severity of fatigue and depression (S3 Table). All other factors not stated were non-contributory.

**Table 2. Multivariable logistic regression analysis of PASC symptoms by acute COVID-19 severity.**

| Symptoms and Outcomes | All participants | Acute COVID-19 Severity | | OR (95% CI) | p-value |
| | | Mild | Severe | | |
| | N = 332 (col %) | N = 171 (col %) | N = 161 (col %) | | |
|---|---|---|---|---|---|
| Dyspnea | 236 (71) | 125 (73) | 110 (69) | 0.98 (0.53–1.80) | 0.32 |
| Fatigue | 199 (60) | 117 (68) | 81 (51) | 1.83 (1.01–3.31)* | 0.047* |
| Cognitive Impairment | 156 (47) | 101 (59) | 55 (34) | 2.76 (1.53–5.00)* | 0.001* |
| Dizziness | 73 (22) | 49 (29) | 24 (15) | 2.41 (1.18–4.92)* | 0.02* |
| Headache | 69 (21) | 48 (28) | 21 (13) | 2.15 (1.05–4.44)* | 0.04* |
| Cough | 98 (30) | 49 (29) | 48 (30) | 1.34 (0.71–2.54) | 0.37 |
| Muscle pain | 61 (18) | 36 (21) | 25 (16) | 0.85 (0.38–1.89) | 0.69 |
| Anxiety | 60 (18) | 40 (23) | 19 (12) | 1.55 (0.73–3.31) | 0.25 |
| Depression | 52 (16) | 24 (14) | 28 (18) | 0.49 (0.22–1.10) | 0.09 |
| Joint pain | 53 (16) | 34 (20) | 19 (12) | 1.12 (0.48–2.59) | 0.78 |
| Palpitations | 53 (16) | 36 (21) | 17 (11) | 1.47 (0.60–3.61) | 0.40 |
| Weakness | 43 (13) | 16 (9) | 27 (17) | 0.66 (0.29–1.51) | 0.32 |
| Sleep disturbances | 40 (12) | 21 (12) | 19 (12) | 0.99 (0.44–2.22) | 0.97 |
| Anosmia | 38 (11) | 21 (17) | 17 (11) | 0.61 (0.23–1.66) | 0.34 |
| Dysgeusia | 50 (15) | 27 (16) | 23 (14) | 0.76 (0.32–1.88) | 0.53 |
| PROMIS Dyspnea >1.5 SD | 9/253 (4) | 1 (<1) | 8 (5) | 0.03 (0.002–0.47)* | 0.01* |
| PROMIS Fatigue >1.5 SD | 75/263 (29) | 47 (27) | 28 (17) | 1.14 (0.55–2.33) | 0.73 |
| PROMIS Cognitive >1.5 SD | 59/259 (23) | 41 (24) | 18 (11) | 2.13 (0.97–4.69) | 0.60 |
| PHQ-9 ≥10 | 84/263 (32) | 45 (26) | 39 (24) | 0.75 (0.44–1.28) | 0.29 |
| GAD-7 ≥10 | 53/127 (42) | 29 (17) | 24 (15) | 1.51 (0.62–2.89) | 0.46 |

*P<0.05 statistically significant

PROMIS = Patient reported outcome measurement information system

SD = Standard deviation

PHQ-9 = Patient health questionnaire-9

GAD-7 = Generalized anxiety disorder-7

## Impact of acute COVID-19 treatment

Acute COVID-19 treatment with remdesivir was associated with less fatigue (OR: 0.47, 95% CI: 0.26–0.86) and fewer reporting severe symptoms (>1.5 SD) on the PROMIS Fatigue (OR: 0.47, 95% CI: 0.23–0.96) and PROMIS Cognitive Function scales (OR: 0.43, 95% CI: 0.20–0.92). This effect was not seen in other symptoms and severity scales.

## Temporal profiles

Fatigue prevalence was highest in those evaluated at 3–6 months from acute COVID-19 onset and persisted over time (Fig 2). This prevalence was highest beyond 3 months (OR: 3.29, 95% CI: 2.08–5.20) (Table 3). Similarly, cognitive impairment prevalence was significantly higher after 3 months (OR: 2.62, 95% CI: 1.67–4.11), but continued to increase over time. Despite this inflection point of 3–6 months for symptom report, PROMIS scales demonstrated that severe fatigue prevalence was highest after 9 months (OR: 3.85, 95% CI: 1.92–7.70) and continued to increase over time, and severe cognitive impairment rates increased significantly at 6–9 months (OR: 2.61, 95% CI: 1.29–5.27) and remained elevated with highest prevalence occurring after 6 months (OR: 2.90, 95% CI: 1.58–5.33). Headaches spiked in prevalence at 9–12 months (OR: 5.80, 95% CI: 1.94–17.3) followed by a decline towards previous prevalence.

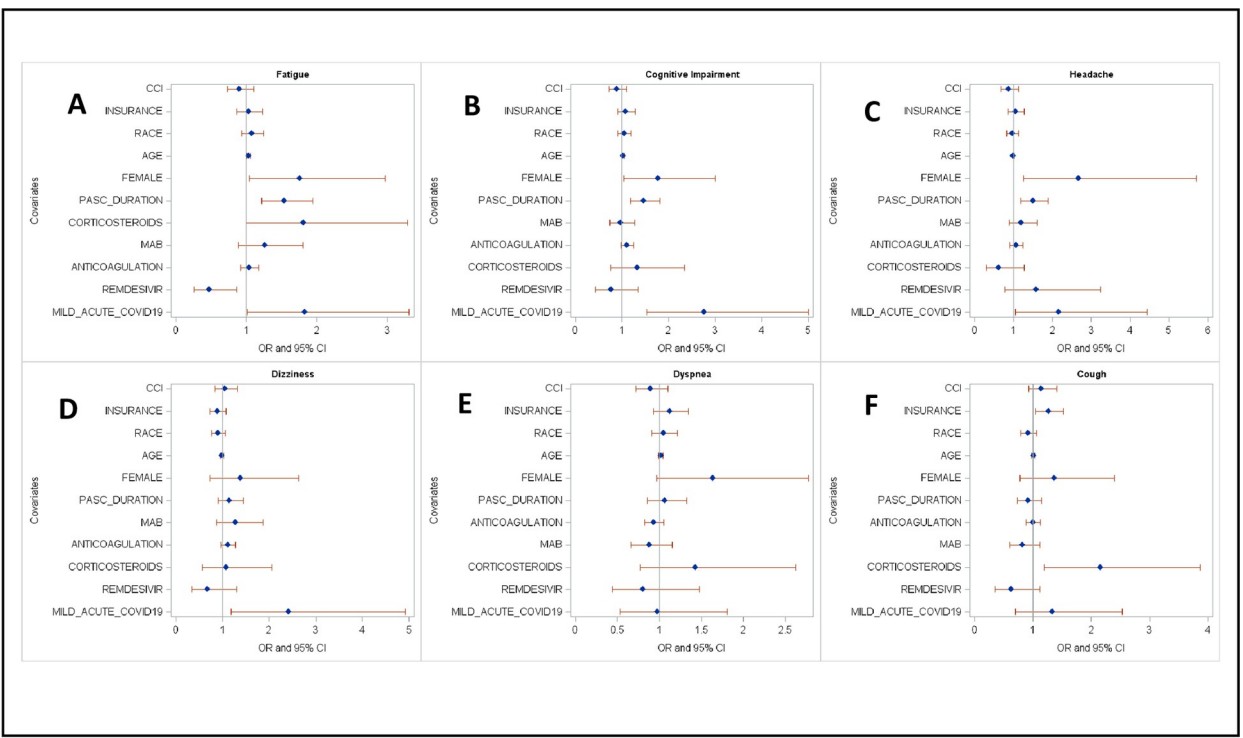

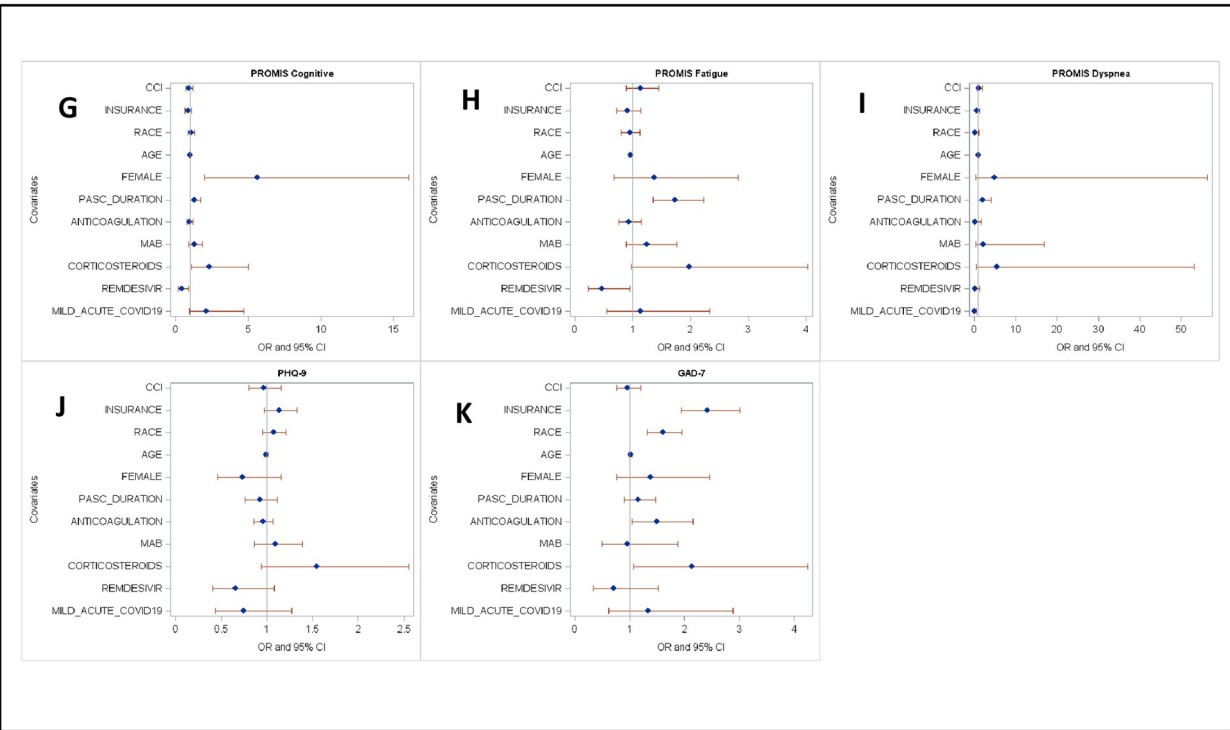

**Fig 1. Factors associated with prevalent PASC symptoms.** Multivariable analysis of factors associated with the most prevalent Post-acute sequelae of SARS-CoV-2 (PASC) symptoms (A) fatigue, (B) subjective cognitive impairment, (C) headache, (D) dizziness, (E) dyspnea, and (F) cough, including Charleston Comorbidity Index (CCI), race, age, sex, PASC duration, acute COVID-19 severity, and acute COVID-19 treatments such as corticosteroids, monoclonal antibodies (MAB), anticoagulation, remdesivir. Sex reference value is female and acute COVID-19 severity reference is mild, as previously defined. PROMIS = Patient reported outcome measurement information system; PHQ-9 = Patient health questionnaire-9; GAD-7 = Generalized anxiety disorder-7.

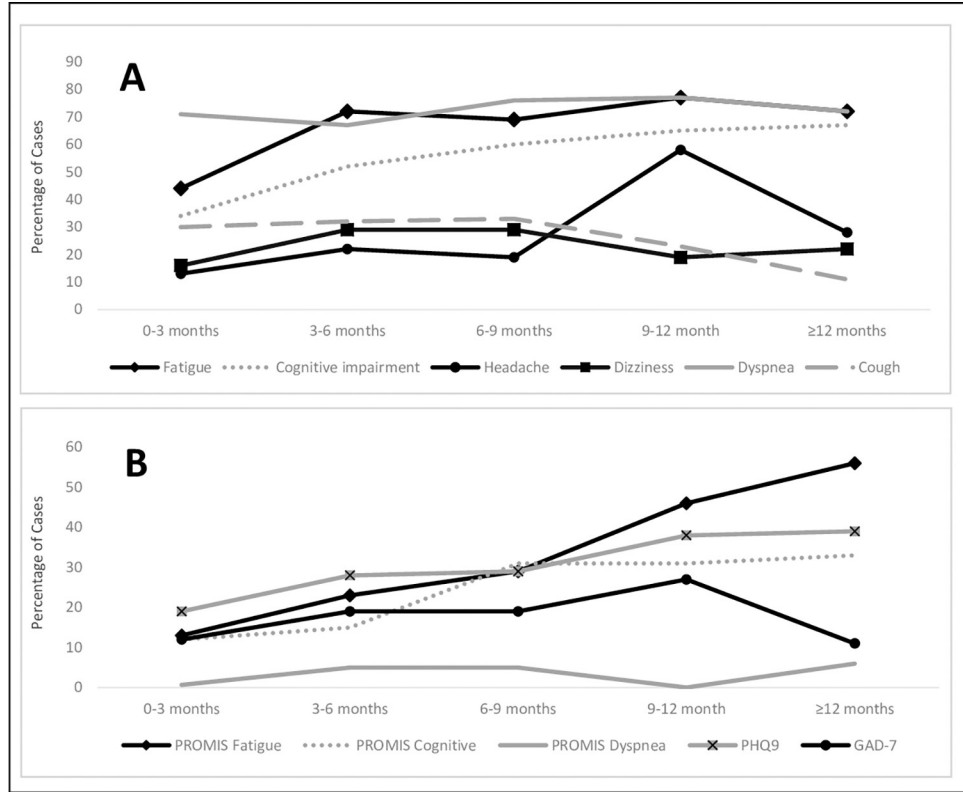

**Fig 2.** Frequency of (A) Prevalent PASC Symptoms and (B) Symptom-specific Severity Outcomes across PASC Duration. (A) Frequency of fatigue, subjective cognitive impairment, headache, dizziness, dyspnea, and cough and (B) Frequency of scores ≥1.5 standard deviations from the mean on PROMIS Fatigue, PROMIS Cognitive, and PROMIS Dyspnea were analyzed across 3-month intervals from incident infection ($T_0$): 0–3, 3–6, 6–9, 9–12, and ≥12 months. PROMIS = Patient reported outcome measurement information system; PHQ-9 = Patient health questionnaire-9; GAD-7 = Generalized anxiety disorder-7; $T_0$ = Time zero.

There were no significant changes in prevalence of other symptoms across PASC durations studied, though dizziness and cough had a trend towards improvement at later time points.

## Discussion

In this study, participants with mild acute COVID-19 and longer PASC durations had increased prevalence of common PASC symptoms. This is a shift in the current theoretical framework that assumes a direct relationship between acute phase severity and PASC development. Although early studies have described higher prevalence of PASC in those with severe acute COVID-19 [14], when controlling for confounders, this association remains equivocal. In previous studies, no association was found in the first 3 months following infection [15, 16], while others found higher prevalence of fatigue, weakness, depression, and anxiety at 6 months following severe acute COVID-19, with no differences in prevalence by 1 year [3, 11]. Vaccination may play role in this discordance, as evidence supports decreased risk of acute COVID-19 severity and decrease risk of PASC development following SARS-CoV-2 vaccination [26, 27]. In contrast to these studies, here we describe higher prevalence of fatigue, cognitive impairment, headaches, and dizziness in those with mild antecedent COVID-19. With mild disease comprising 94% of COVID-19 cases and PASC occurring in a substantial proportion of survivors, our findings have important policy implications [12, 28, 29]. Vaccination

**Table 3. PASC duration intervals associated with prevalent symptoms and symptom-specific severity outcomes.**

| Symptoms and Outcomes | PASC Duration Intervals, months | | | | | |
|---|---|---|---|---|---|---|
| | 3–6 vs 0–3 | 6–9 vs 3–6 | 9–12 vs 6–9 | 3–12+ vs 0–3 | 6–12+ vs 0–6 | 9–12+ vs 0–9 |
| | OR (95% CI) | OR (95% CI) | OR (95% CI) | OR (95% CI) | OR (95% CI) | OR (95% CI) |
| Symptoms | | | | | | |
| Fatigue | 3.30 (1.91–5.69)* | 0.86 (0.39–1.88) | 1.49 (0.49–4.59) | 3.29 (2.08–5.20)* | 2.06 (1.20–3.51)* | 2.21 (1.07–4.54)* |
| Cognitive Impairment | 2.16 (1.29–3.64) | 1.33 (0.64–2.76) | 1.28 (0.47–3.55) | 2.62 (1.67–4.11)* | 2.38 (1.44–3.95)** | 2.45 (1.26–4.77)* |
| Dizziness | 2.14 (1.15–3.97) | 0.99 (0.45–2.20) | 0.60 (0.18–1.94) | 1.94 (1.11–3.36)* | 1.21 (0.68–2.15) | 0.90 (0.41–1.97) |
| Headache | 1.85 (0.94–3.63) | 0.34 (0.34–2.09) | 5.80 (1.94–17.3)* | 2.42 (1.35–4.33)* | 2.41 (1.38–4.23)* | 4.07 (2.08–7.93)* |
| Dyspnea | 0.84 (0.49–1.46) | 1.55 (0.68–3.54) | 1.04 (0.33–3.31) | 1.00 (0.62–1.62) | 1.36 (0.77–2.38) | 1.26 (0.61–2.60) |
| Cough | 1.07 (0.62–1.84) | 1.08 (0.50–2.32) | 0.60 (0.20–1.83) | 0.93 (0.58–1.50) | 0.77 (0.44–1.34) | 0.49 (0.22–1.09) |
| Symptom Severity Outcomes | | | | | | |
| PROMIS Dyspnea >1.5 SD | 0.70 (0.39–1.24) | 1.64 (0.75–3.57) | 0.38 (0.11–1.30) | 5.71 (0.70–46.3) | 1.36 (0.33–5.58) | 0.68 (0.08–5.60) |
| PROMIS Fatigue >1.5 SD | 0.77 (0.46–1.29) | 1.82 (0.91–3.62) | 0.92 (0.37–2.31) | 2.58 (1.43–4.68)* | 3.20 (1.80–5.68)* | 3.85 (1.92–7.70)* |
| PROMIS Cognitive >1.5 SD | 0.60 (0.35–1.02) | 2.61 (1.29–5.27)* | 0.50 (0.19–1.29) | 1.76 (0.95–3.27) | 2.90 (1.58–5.33)* | 2.04 (0.99–4.21) |
| PHQ-9 ≥10 | 0.68 (0.42–1.12) | 1.68 (0.83–3.27) | 0.75 (0.30–1.90) | 0.77 (0.51–1.17) | 1.06 (0.66–1.69) | 0.86 (0.46–1.59) |
| GAD-7 ≥10 | 1.14 (0.69–1.89) | 1.03 (0.50–2.11) | 0.78 (0.30–2.02) | 1.18 (0.77–1.82) | 1.17 (0.71–1.92) | 1.21 (0.63–2.30) |

*P<0.05 statistically significant

PROMIS = Patient reported outcome measurement information system

SD = Standard deviation

PHQ-9 = Patient health questionnaire-9

GAD-7 = Generalized anxiety disorder-7

campaigns and masking strategies primarily focus on mitigating risk of acute COVID-19 morbidity and mortality, but our findings underscore the importance of preventing mild disease given these known, debilitating sequelae. This effect may have gone undetected previously as few studies have analyzed risk factors associated with independent PASC symptoms [3, 13, 15, 16]. Interestingly, dyspnea severity had an opposite association wherein it was directly correlated with severe acute COVID-19, and report of any dyspnea was associated with critical incident disease. Further investigation is needed to determine if distinct phenotypes exist within PASC that prevent analysis of symptoms in aggregate.

Our study design has the advantage of characterizing symptom profiles at multiple PASC intervals, including longer intervals which have not been well characterized. Complaints of fatigue, cognitive impairment, and headache occurred at higher prevalence in those with longer PASC duration. In fact, a PASC duration of 3–6 months appears to be a breakpoint at which fatigue becomes more prevalent and persists in our population. Similarly, prevalence of subjective cognitive impairment rapidly increases in those at 3–6 months from incident infection, followed by a steady increase in those with longer PASC durations (>6 months). Although there is still much to learn about PASC symptom clusters, this supports demarcation of a distinct, delayed phenotype that, for some, may not improve with time. Interestingly, our study showed an association with remdesivir, an anti-viral which may provide some protection against these symptoms. This delayed phenotype overlaps with a symptom cluster and longitudinal course described previously [6], but differs from others who describe higher fatigue beyond one year in PASC with no change in cognitive symptoms over time [13]. Additionally, we found that prevalence of fatigue and cognitive impairment severity continues to increase in participants with PASC of longer durations, with severity escalating after 9 months for fatigue and after 6 months for cognitive impairment [13]. Characterization of cognitive impairment and severity is key to this phenotype and is an important element of this study as

neurocognitive effects are among the most prevalent PASC symptoms but are-understudied in the current literature [3, 4, 6, 11, 16, 30]. The temporal pattern for headaches appears unique from fatigue and cognitive impairment with headache prevalence peaking at 9–12 months, followed by quick decline. Dizziness prevalence increased and remained elevated in those 3–9 months post incident SARS-CoV-2 infection followed by improvement later in PASC course, consistent with previous reports [6]. Similarly, prevalence of cough declined over time. Although prevalence of dyspnea remains static, severity of symptoms may continue to increase over time, consistent with previous findings showing higher prevalence at 12 months [13].

In our study, receipt of remdesivir during the acute phase was associated with less fatigue and decreased severity of fatigue and subjective cognitive impairment. Antivirals have not shown benefit previously; however, prior analyses investigated composite antivirals, including drugs with diverse mechanisms such as hydroxychloroquine, which has shown no significant effect in the acute COVID-19 outcomes [3, 11, 31]. The mechanism by which remdesivir may impact fatigue and cognitive impairment is unclear. Though studies have failed to show significant nasopharyngeal viral load reduction in SARS-CoV-2 infected individuals following remdesivir treatment [32], the potential protective effects against this symptom profile may support the theory of a persistent viral reservoir that triggers ongoing immune responses, contributing to the underlying pathology of PASC [33, 34]. Further studies are needed to determine if remdesivir interrupts this cascade through viral clearance in other tissues that are not accurately represented by nasopharyngeal sampling. Perhaps the use of anti-viral treatments in COVID-19 is reminiscent of antimicrobial treatment in group A streptococcus infections, wherein early treatment provides protection against post-acute sequelae affecting multiple organ systems [35]. Remdesivir use showed no association with development or persistence of dyspnea or cough, and there was no significant temporal pattern for these symptoms, which may indicate a heterogenous underlying mechanism in those experiencing these symptoms. This is consistent with the previously described cluster phenotype, which excluded pulmonary symptoms [6]. Corticosteroid administration was associated with persistent cough and the etiology of the association remains unclear. It may represent increased requirement of corticosteroid treatment among those with chronic respiratory disease during COVID-19 infection in conjunction with under-ascertainment of chronic respiratory disease, as is common in this population [36–38].

Female PASC patients were more prone to fatigue, cognitive impairment, and headaches than male counterparts. Women have been reported to have higher rates of composite PASC symptoms [30], though, adjusted models have found either no association [16] or symptom-specific associations with fatigue, headache, anxiety and depression [3, 28]. In our analysis, those with advanced age had more fatigue. Advanced age has been identified as a risk factor for PASC development [14, 30], with higher risk of fatigue, cognitive impairments, weakness, and joint pain in adjusted models [3, 7, 39]. However, findings are inconsistent, with some studies showing no impact of age on individual symptoms [16, 17]. Similarly, minority race has been shown to contribute higher morbidity and mortality from acute COVID-19 [40] and recent studies support higher incidence of long-term sequelae including diabetes, pulmonary embolism, chest pain, cough, and headaches in African Americans and headaches, dyspnea, and joint and chest pain in Hispanic patients [39, 41]. However, race has not remained significant in adjusted PASC models for symptoms such as fatigue, memory problems, or sleep impairments, and may occur more frequently in white subjects [17, 39, 41]. Our study included a diverse population with broad socioeconomic representation, and we only observed a sociodemographic effect for white, privately insured participants who experienced more severe anxiety.

There are several limitations of this study. As this study exclusively enrolled PASC patients presenting to care, symptom prevalence is overestimated compared with the general

population. Additionally, this design is subject to referral bias and may skew towards a PASC population with higher symptom burden and severity. Reported symptom percentages represent symptom burden in a population of PASC patients referred for specialty care and are presented to demonstrate relative proportions between groups to study contributory and mitigating factors associated with symptom profiles. Incomplete availability of vaccination data prevented inclusion of vaccination status in our multivariable models and may bias our results as evidence supports decreased acute COVID-19 severity and PASC development following SARS-CoV-2 vaccination [26, 27]. Limiting enrollment to PASC clinic patients risks exclusion of historically marginalized groups including those with financial limitations and poor access to care; however, 34% of enrollees were uninsured or Medicaid recipients. During the study period, our PASC clinic referral queue exceeded our ability to schedule standardized follow-up visits and many patients were lost to follow-up. As such our study does not provide longitudinal assessments with multiple PASC duration time points per patient. Rather, our PASC duration groups represent distinct patient populations. Participant characteristics did not differ significantly between PASC duration groups, with the exception of higher proportion of black race in those with shorter PASC duration; however, race did not contribute significantly in adjusted models. Due to the study period, Alpha was the predominant circulating SARS-CoV-2 strain, limiting evaluation of each variant's impact on symptom profiles, including the Omicron variant. Given the observational nature of this study, the associations described between remdesivir use and decreased fatigue and cognitive impairment should be interpreted with caution as there may be additional confounding factors not accounted for in the study design. To better inform this relationship, remdesivir could be considered for future randomized clinical trials.

In this study, we demonstrate that mild acute COVID-19 and PASC duration are associated with fatigue, subjective cognitive impairment, headaches, and dizziness and that those treated with remdesivir in the acute phase experienced lower prevalence and severity of fatigue as well as less severity of cognitive impairment. Though the pathobiology of PASC remains unknown, these findings provide some insight into potential physiologic mechanisms and targets for prevention of particular PASC phenotypes. Furthermore, we show that these symptoms may have a delayed onset ranging 3–12 months from incident infection, and a subset of these patients will continue to have persistent symptoms that do not improve significantly over time, underscoring the importance for targeted COVID-19 preventative measures.

## Supporting information

**S1 Table. Post-acute sequelae of SARS-CoV-2 review of symptoms.**
(DOCX)

**S2 Table. Multivariable logistic regression analysis of PASC symptoms by COVID-19 variant–Alpha vs Delta variant.**
(DOCX)

**S3 Table. Multivariable logistic regression analysis of PASC symptoms by acute COVID-19 severity–Critical vs Non-critical.**
(DOCX)

## Author Contributions

**Conceptualization:** Tiffany A. Walker, Alex D. Truong, Aerica Summers, Felicia C. Goldstein, Ihab Hajjar, Melvin R. Echols, Erica D. Lee, Seema Tekwani, Kelley Carroll, Ignacio Sanz, F. Eun-Hyung Lee, Jenny E. Han.

**Data curation:** Tiffany A. Walker, F. Eun-Hyung Lee, Jenny E. Han.

**Formal analysis:** Tiffany A. Walker.

**Funding acquisition:** Tiffany A. Walker, Melvin R. Echols, Kelley Carroll, Ignacio Sanz, F. Eun-Hyung Lee, Jenny E. Han.

**Investigation:** Tiffany A. Walker, Alex D. Truong, Aerica Summers, Adviteeya N. Dixit, F. Eun-Hyung Lee, Jenny E. Han.

**Methodology:** Tiffany A. Walker, Alex D. Truong, Aerica Summers, Felicia C. Goldstein, Ihab Hajjar, Melvin R. Echols, Erica D. Lee, Seema Tekwani, Kelley Carroll, Ignacio Sanz, F. Eun-Hyung Lee, Jenny E. Han.

**Project administration:** Tiffany A. Walker, Alex D. Truong, Aerica Summers, Adviteeya N. Dixit, Kelley Carroll, F. Eun-Hyung Lee, Jenny E. Han.

**Resources:** Felicia C. Goldstein, Ihab Hajjar, Melvin R. Echols, Matthew C. Woodruff, Erica D. Lee, Kelley Carroll, F. Eun-Hyung Lee, Jenny E. Han.

**Supervision:** Tiffany A. Walker, Felicia C. Goldstein, Ihab Hajjar, F. Eun-Hyung Lee, Jenny E. Han.

**Validation:** Tiffany A. Walker, Jenny E. Han.

**Visualization:** Tiffany A. Walker.

**Writing – original draft:** Tiffany A. Walker.

**Writing – review & editing:** Alex D. Truong, Aerica Summers, Adviteeya N. Dixit, Felicia C. Goldstein, Ihab Hajjar, Melvin R. Echols, Matthew C. Woodruff, Erica D. Lee, Seema Tekwani, Kelley Carroll, Ignacio Sanz, F. Eun-Hyung Lee, Jenny E. Han.

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
