## [Decision Letter · Decision Letter 0]

16 Mar 2023

PONE-D-22-27140Mild antecedent COVID-19 associated with symptom-specific post-acute sequelaePLOS ONE

Dear Dr. Walker,

Thank you for submitting your manuscript to PLOS ONE. After careful consideration, we feel that it has merit but does not fully meet PLOS ONE’s publication criteria as it currently stands. Therefore, we invite you to submit a revised version of the manuscript that addresses the points raised during the review process.

We look forward to receiving your revised manuscript.

Kind regards,

Jung Yeon Heo

Academic Editor

PLOS ONE

2. Thank you for stating the following in the Acknowledgments/ Funding Section of your manuscript:

“This work was supported by the Woodruff Health Science Center COVID-19 CURE Award through philanthropic support from the O. Wayne Rollins Foundation and the William Randolph Hearst Foundation and through in-kind support from Grady Healthcare. The funding source has no role in study design, collection, analysis or interpretation of data, writing of reports, nor decision to submit papers for publication. The findings and conclusions in this report are those of the authors and do not necessarily represent the official position of the Woodruff Health Science Center, Emory University, Morehouse School of Medicine, Grady Memorial Hospital, or affiliated partners.”

“TAW and JEH were funded by the Woodruff Health Sciences Center COVID-19 CURE Award. There is no associated award number. http://whsc.emory.edu/research/covid-19-research/index.html

“I have read the journal's policy and the authors of this manuscript have the following competing interests: Dr. F. Eun-Hyung Lee reports grants from Genetech and the Gates Foundation. She has received royalties for BLI, Inc and consulting fees from Be Bio Pharma. She received honoraria from Gerontological Advanced Practice Nurses Association and has patents for plasma cell survival media and MENSA. She is the founder of the MicroB-plex, Inc. Dr. Ignacio Sanz reports grants from GlaxoSmithKline, Bristol Myers Squibb, Exagen, and consulting fees from Pfizer, Octagon, and Bristol Myers Squibb. He serves on the DSMB for and has stock options in Kyverna.”

Reviewers' comments:

Reviewer's Responses to Questions

**Comments to the Author**

1. Is the manuscript technically sound, and do the data support the conclusions?

Reviewer #1: Yes

Reviewer #2: Yes

2. Has the statistical analysis been performed appropriately and rigorously? 

Reviewer #1: Yes

Reviewer #2: Yes

3. Have the authors made all data underlying the findings in their manuscript fully available?

Reviewer #1: Yes

Reviewer #2: Yes

4. Is the manuscript presented in an intelligible fashion and written in standard English?

Reviewer #1: Yes

Reviewer #2: Yes

5. Review Comments to the Author

Reviewer #1: This manuscript presents novel evidence to our understanding of long COVID and development of post-acute sequelae among individuals with mild acute symptoms. I had a number of minor edits that I detail in the attached document, but I believe that after addressing my comments this paper should add key knowledge to the scientific literature on long COVID.

I would especially urge the authors to use prevalence instead of rate throughout the manuscript as to not confuse readers who may not understand the difference between prevalence and incidence.

Best,

Reviewer

Reviewer #2: This study analyzed the risk for each symptom of PASC, and also demonstrated the prevalence of symptom profiles over time to evaluate the disease course. Authors dealt with an interesting topic and drew valid conclusions through a thorough literature search and data analysis. However, I think some revisions are required.

<major>

1. Variant type of SARS-CoV-2 were changed during the study period. Didn't you check the effect of the variant? Even if the virus type cannot be determined through the PCR test, the analysis can be performed based on the time point when the omicron became the dominant variant.

2. In this study, severe patients were not separated from critical patients. It would be desirable to demonstrate the PASC of critical patients by subgroup analysis.

3. In Table 1, mild patients were persons who were not hospitalized, but how did they receive remdesivir. Did all these patients receive it outpatients?

4. Since vaccination is an important factor that can affect the PASC, authors should describe the point that effect of vaccine was not analyzed as a limitation in the discussion section.

<minor>

1. Why do you think dexamethasone was a risk factor for persistent cough after COVID-19?</minor></major>

6. PLOS authors have the option to publish the peer review history of their article (what does this mean?). If published, this will include your full peer review and any attached files.

Reviewer #1: No

Reviewer #2: No

---

## [Author Response · Author response to Decision Letter 0]

11 Apr 2023

Manuscript Text Reviewer Comments: 

1. Depending on time of publishing may need to update these numbers

a) Author response: Updated with current case and death counts in Line 51

2. Many of the studies you cite are among hospitalized patients – the prevalence of PASC among all individuals infected with SARS-CoV-2 is much lower so I would update this value

https://pubmed.ncbi.nlm.nih.gov/34007978/

https://academic.oup.com/cid/article/73/11/2055/6276644

https://academic.oup.com/jid/article/226/9/1593/6569364?login=false

a. Author response: I agree with this point. Included these citations as well as the Bull-Otterson MMWR: https://www.cdc.gov/mmwr/volumes/71/wr/mm7121e1.htm in Line 56

3. Are you distinguishing between symptoms that occur during acute infection that last for 3+ months and new symptoms that occur after initial infection? I see you mention persisting symptoms later in methods – feel free to disregard

a. Author response: no edits suggested

4. Public health education or clinical education? It seems that the target audience of this paper is more clinical rather than geared towards public health practitioners

a. Author response: Good point. I have made this suggested edit in Line 59

5. Concurrently?

a. Author response: I chose consecutively to demonstrate that all patients presenting to our PASC clinics had an opportunity to enroll. We did not skip patients or select certain patients. I'm concerned that "concurrently" doesn't indicate this equity in sampling, but I am open to the change if the editor agrees with the suggested edit.

6. Should maybe mention how vaccination can potentially affect risk of PASC in discussion. Would potentially be a confounder for symptom severity and PASC outcome

a. Author response: Good point. I have added a comment on vaccination impact on acute disease severity and PASC development to the discussion, Line 210 and limitations, Line 286. 

7. I would make this consistent with your methods above, it was a little unclear to me in the earlier section

a. Author response: Thank you. Edit addressed in Line 85.

8. For multivariable models how were parameters selected for inclusion? What metrics (ie AIC) were used to determine final model structure?

a. Author response: Variables that were significant on univariate analysis and those with clinical relevance were incorporated into the multivariable model. AIC was used to determine goodness-of-fit, and variables optimizing AIC were retained. Edit addressed in the text in Line 121.

9. Acute symptom clinical variables? Clinical variables from enrollment at the PASC center?

a. Author response: Clinical variables include: sociodemographic, comorbidity, acute COVID-19 severity and treatment variables. Edit is addressed in the text on Line 119.

10. Citation for software?

a. Author response: Reference was added to the text, Line 125

11. Did you perform a sample size calculation?

a. Author response: A sample size calculation was performed prior to study initiation, estimating 300 participants needed to achieve 90% power to detect changes in sequelae trajectories in longitudinal analysis. However, we were unable to attain repeated measures in a sufficient number of patients due to burden of new PASC patients filling appointments in our PASC clinics leaving limited appointments for follow-up patients, as described in the study limitations section. For this reason, we were unable to analyze longitudinal trends in symptomatology over time. 

12. Did you combine race and ethnicity into one category? Please clarify in methods.

a. Author response: Race and ethnicity were analyzed together as one variable in models. Edit is addressed in the text, Line 120

13. These sentences could potentially be combined, but not necessary if authors prefer to keep separate

a. Author response: I agree. Edit is addressed in the text, Line 137 

14. This should be months after incident infection – the current wording could indicate that participants only experience PASC for X duration. Changing wording to reflect time elapsed (months after incidence infection) can help make it clear that participants are experiencing PASC for at least X amount of time

a. Author response: I have made this suggested edit in Table 1

15. Would be good to write out just so it’s easier for the reader to understand values at a glance

a. Author response: Agree with edit made in the text in Table 1

16. What do you categorize as longer PASC?

a. Author response: PASC duration was quantified as months from incident COVID-19 onset and analyzed in 3-month intervals (0-3, 3-6, 6-9, 9-12, ≥12 months). In models, PASC duration was positively correlated with fatigue, cognitive impairment, headaches, severe fatigue, and subjective cognitive impairment, indicating that those with longer PASC durations (those at the longer end of the spectrum) have higher odds of these symptoms. Clarification made in the text, Line 154

17. Reiterate that these are percentages

a. Author response: I have made this suggested edit in Table 2.

18. I believe PLOS asks for 2 significant figures so double check all tables

a. Author response: I have not seen guidance on significant digits, but I can revise if suggested by the editor. I have corrected the error noted in the text in Table 2.

19. What are you defining as advanced age?

a. Author response: Updated the text to define advanced age as (≥65 years) in Line 170

20. Please clarify use of rate versus prevalence throughout. If you are only capturing new cases of PASC it is fine to use rate in reference to incidence but otherwise prevalence should be used. Since you used prevalence in L173 please use it consistently

a. Author response: Updated the text to change “rate” to prevalence at all points. 

21. Revise if this language is incorrect

a. Author response: Text has been updated to use the language “prevalence”. No further edits needed.

22. Please use prevalence instead of rate throughout

a. Author response: Edits made in the text as requested

23. Comment on how vaccination status can affect acute symptom severity and potentially PASC development

a. Author response: I have added a comment on vaccination impact on acute disease severity and PASC development to the discussion, Line 210 and limitations, Line 286. 

24. This sentences seems to imply tracking patients repeatedly but you indicated in methods that your study participants were only evaluated once. Instead of saying over time, switch focus to increase by time since incident infection

a. Author response: Agree with this point as well. Please see amended text, Line 227

25. Same comment as before regarding time

a. Author response: Agree with this point as well. Please see amended text, Line 234

26. I would double check the epidemiologic literature for this

a. Author response: Please see suggested edits. I updated the text with current literature on age as a risk factor for individual PASC symptoms, Line 269

27. Please check some of these other epidemiological studies that do show an association between race/ethnicity and PASC.

https://www.ncbi.nlm.nih.gov/pmc/articles/PMC8445372/

https://www.nature.com/articles/s41467-021-26513-3

a. Author response: Thank you. When individual symptoms are analyzed, it appears that there may be symptoms that are more common in specific racial/ethnic groups. Many of the same symptoms we studied remained insignificant between racial groups in adjusted models in the Xie paper referenced (see Xie Supplementary Table 6), and some of these symptoms were found to be more common in white participants in the Khuller adjusted analysis. See suggested edits in the text, Line 274

Reviewer 1 General comments: 

Reviewer #1: This manuscript presents novel evidence to our understanding of long COVID and development of post-acute sequelae among individuals with mild acute symptoms. I had a number of minor edits that I detail in the attached document, but I believe that after addressing my comments this paper should add key knowledge to the scientific literature on long COVID.

I would especially urge the authors to use prevalence instead of rate throughout the manuscript as to not confuse readers who may not understand the difference between prevalence and incidence.

a. Author response: Thank you for this suggestion. I have updated the text with this correction. 

Reviewer 2 General comments:

Reviewer #2: This study analyzed the risk for each symptom of PASC, and also demonstrated the prevalence of symptom profiles over time to evaluate the disease course. Authors dealt with an interesting topic and drew valid conclusions through a thorough literature search and data analysis. However, I think some revisions are required.

1. Variant type of SARS-CoV-2 were changed during the study period. Didn't you check the effect of the variant? Even if the virus type cannot be determined through the PCR test, the analysis can be performed based on the time point when the omicron became the dominant variant.

a. Author response: The authors agree with this suggestion. Given the time span that the study was conducted, the majority (94%) of the infections were the original strain and the remaining 6% were presumed Delta (infections after 7/1/2021 and before 12/1/2021). There were no infections after 12/1/2021 in our cohort. We have provided a subgroup analysis as Appendix II demonstrating the effect of variant on symptom type. Most are not statistically significant, likely due to the small sample size for Delta; however, we did see less anxiety in alpha (more anxiety in Delta). The text has been updated: methods described definition (Line 88) of variants and subgroup analysis (Line 118), results reference Appendix II findings (136), discussion mentions limitation of not having more diversity of variants to evaluate effect of currently circulating strain (Line 297). 

2. In this study, severe patients were not separated from critical patients. It would be desirable to demonstrate the PASC of critical patients by subgroup analysis.

a. Author response: Thank you for this suggestion. We conducted a subgroup analysis of critical acute COVID-19 patients and found that critical COVID-19 is associated with more dyspnea and less subjective cognitive impairment, joint pain, anosmia, and severity of fatigue and depression, consistent with general trends seen in the mild vs severe COVID-19 comparison. This population only comprises 12% of the study population and small sample size may affect ability to detect differences in other variables. The text has been updated in the results section, referencing the findings (Appendix III)- Line 175, and the discussion has been updated, Line 220. The methods section already defines critical COVID-19. 

3. In Table 1, mild patients were persons who were not hospitalized, but how did they receive remdesivir. Did all these patients receive it outpatients?

a. Author response: Four patients received 3 days of outpatient infusions at an infusion center. This was off label and at the discretion of their provider. I have expanded the explanation in the methods section, Line 98

4. Since vaccination is an important factor that can affect the PASC, authors should describe the point that effect of vaccine was not analyzed as a limitation in the discussion section.

Author response: I have added a comment on vaccination impact on acute disease severity and PASC development to the discussion, Line 210 and limitations, Line 286.

1. Why do you think dexamethasone was a risk factor for persistent cough after COVID-19?

a. Author response: This is unclear. It could be due to the fact that those with chronic pulmonary diseases (COPD, asthma) were more likely to require dexamethasone treatment during a COVID-19 infection. The adjusted model analyzes comorbidities by the Charleston Comorbidity Index and not individual comorbidities. However, when I conduct a sensitivity analysis using COPD and asthma in place of the CCI, there remains a slight effect for dexamethasone and persistent cough. Potentially, there are participants with underlying pulmonary diseases that have gone undiagnosed, driving this effect. Explanation is added to the discussion, Line 263.

---

## [Decision Letter · Decision Letter 1]

2 May 2023

PONE-D-22-27140R1Mild antecedent COVID-19 associated with symptom-specific post-acute sequelaePLOS ONE

Dear Dr. Walker,

Thank you for submitting your manuscript to PLOS ONE. After careful consideration, we feel that it has merit but does not fully meet PLOS ONE’s publication criteria as it currently stands. Therefore, we invite you to submit a revised version of the manuscript that addresses the points raised during the review process.

We look forward to receiving your revised manuscript.

Kind regards,

Jung Yeon Heo

Academic Editor

PLOS ONE

Journal Requirements:

Additional Editor Comments:

We really appreciate your efforts to improve the patient management for COVID-19 and to get scientific evidence more. However, some points need to be still revised before publication to raise completeness.

- The authors should include their description of the sample size calculation in the methods section. It was noted in response to reviewer’s comments

- All abbreviation should initially spell out in main text and table/figure respectively. e.g. AIC in main text and AI, AA, CCI in table, and so on

- All tables and figures are showed in main text. However, I can’t find table 2 in main text.

- The authors used 3 different supplementary materials. However, the authors used same number, supplementary appendix II, twice in table title. To avoid confusion, I suggest to use supplementary table 1, 2 and 3 in main text and title instead of appendix I and II

- in line 154-158, In which table or figure is this description presented?

- Is it right for figure 2 title and legend? Please add a title of Y axis in table 2

Reviewers' comments:

Reviewer's Responses to Questions

**Comments to the Author**

1. If the authors have adequately addressed your comments raised in a previous round of review and you feel that this manuscript is now acceptable for publication, you may indicate that here to bypass the “Comments to the Author” section, enter your conflict of interest statement in the “Confidential to Editor” section, and submit your "Accept" recommendation.

Reviewer #1: All comments have been addressed

Reviewer #2: All comments have been addressed

2. Is the manuscript technically sound, and do the data support the conclusions?

Reviewer #1: Yes

Reviewer #2: Yes

3. Has the statistical analysis been performed appropriately and rigorously? 

Reviewer #1: Yes

Reviewer #2: Yes

4. Have the authors made all data underlying the findings in their manuscript fully available?

Reviewer #1: Yes

Reviewer #2: Yes

5. Is the manuscript presented in an intelligible fashion and written in standard English?

Reviewer #1: Yes

Reviewer #2: Yes

6. Review Comments to the Author

Reviewer #1: Great job on revisions and clarity - my only comment remaining is that the authors should include their description of the sample size calculation in the methods section which was provided in the last round of rebuttal ("A sample size calculation...estimating 300 participants...analyze longitudinal trends over time").

After these details are added this manuscript is ready for publication.

Reviewer #2: (No Response)

7. PLOS authors have the option to publish the peer review history of their article (what does this mean?). If published, this will include your full peer review and any attached files.

Reviewer #1: No

Reviewer #2: No

---

## [Author Response · Author response to Decision Letter 1]

13 Jun 2023

1. The authors should include their description of the sample size calculation in the methods section. It was noted in response to reviewer’s comments

Author response: Edits made. Power calculation added to Statistical analysis Line 117*. Limitations in completing longitudinal analysis addressed previously in Line 309*

2. All abbreviation should initially spell out in main text and table/figure respectively. e.g. AIC in main text and AI, AA, CCI in table, and so on

Author response: Edits made. Abbreviations/acronyms have been spelled out in text, tables, and figures. 

3. All tables and figures are showed in main text. However, I can’t find table 2 in main text.

Author response: Table 2 is present in Line 165* and referenced on Line 151-152*

4. The authors used 3 different supplementary materials. However, the authors used same number, supplementary appendix II, twice in table title. To avoid confusion, I suggest to use supplementary table 1, 2 and 3 in main text and title instead of appendix I and II

Author response: Edit made. All supplementary materials are now listed as Supplementary Table 1, 2, and 3 in the main text and supplement title. 

5. In line 154-158, In which table or figure is this description presented?

Author response: All text in this paragraph is in reference to Figure 1, which is referenced in Line 158*. I have added an additional reference to Figure 1 in Line 160* for clarity. 

6. Is it right for figure 2 title and legend? Please add a title of Y axis in table 2

Author response: Yes, the Figure 2 title and legend are correct. Title of Y axis has been added to table 2. 

*Listed Line numbers refers to location in tracked-changes version

---

## [Editor Report · Decision Letter 2]

15 Jun 2023

PONE-D-22-27140R2Mild antecedent COVID-19 associated with symptom-specific post-acute sequelaePLOS ONE

Dear Dr. Walker,

Thank you for submitting your manuscript to PLOS ONE. After careful consideration, we feel that it has merit but does not fully meet PLOS ONE’s publication criteria as it currently stands. Therefore, we invite you to submit a revised version of the manuscript that addresses the points raised during the review process.

We look forward to receiving your revised manuscript.

Kind regards,

Jung Yeon Heo

Academic Editor

PLOS ONE

Journal Requirements:

Additional Editor Comments :

Thank you for submitting the revised manuscript.

We really appreciate your efforts to improve the patient management for COVID-19 and to get scientific evidence more. However, some points need to be checked before publication to raise completeness.

- In line 187-190, the association between remdesivir and PASC symptoms is a key point in this paper. However, I cannot find in which table or figure this description was presented. Please check it again.

- It would be better to avoid repeating figure or table cited in main text. So, Figure 1 can be removed presenting in line 159.

---

## [Author Response · Author response to Decision Letter 2]

16 Jun 2023

1. In line 187-190, the association between remdesivir and PASC symptoms is a key point in this paper. However, I cannot find in which table or figure this description was presented. Please check it again.

a. Author response: These data are represented in Figure 1 and follow a paragraph that discusses data represented in Figure 1. I have not added additional reference to Figure 1 per your recommendation below, but would be happy to do so if needed for clarity. 

2. It would be better to avoid repeating figure or table cited in main text. So, Figure 1 can be removed presenting in line 159.

a. Author response: Duplicate reference to Figure 1 has been removed.

---

## [Editor Report · Decision Letter 3]

26 Jun 2023

Mild antecedent COVID-19 associated with symptom-specific post-acute sequelae

PONE-D-22-27140R3

Dear Dr. Walker,

We’re pleased to inform you that your manuscript has been judged scientifically suitable for publication and will be formally accepted for publication once it meets all outstanding technical requirements.

Kind regards,

Jung Yeon Heo

Academic Editor

PLOS ONE
---

## [Editor Report · Acceptance letter]

29 Jun 2023

PONE-D-22-27140R3 

Mild antecedent COVID-19 associated with symptom-specific post-acute sequelae 

Dear Dr. Walker:

I'm pleased to inform you that your manuscript has been deemed suitable for publication in PLOS ONE. Congratulations! Your manuscript is now with our production department. 

Kind regards, 

on behalf of

Dr. Jung Yeon Heo 

Academic Editor

PLOS ONE